# Vasovagal Syncope Is Associated with Variants in Genes Involved in Neurohumoral Signaling Pathways

**DOI:** 10.3390/genes13091653

**Published:** 2022-09-15

**Authors:** Boris Titov, Natalya Matveeva, Olga Kulakova, Natalia Baulina, Elizaveta Bazyleva, Grigory Kheymets, Anatolii Rogoza, Alexander Pevzner, Olga Favorova

**Affiliations:** 1Chazov National Medical Research Center of Cardiology, 121552 Moscow, Russia; 2Laboratory of Medical Genomics, Pirogov Russian National Research Medical University, 117997 Moscow, Russia

**Keywords:** syncope, vasovagal syncope, genetic polymorphism, genetic predisposition, signal transduction

## Abstract

Vasovagal syncope (VVS) is the most common cause of sudden loss of consciousness. VVS results from cerebral hypoperfusion, due to abnormal autonomic control of blood circulation, leading to arterial hypotension. It is a complex disease, and its development is largely associated with genetic susceptibility. Since abnormal neurohumoral regulation plays an important role in VVS development, we analyzed the association of VVS with polymorphic variants of *ADRA1A*, *ADRB1*, *HTR1A*, *ADORA2A*, *COMT*, and *NOS3* genes, the products of which are involved in neurohumoral signaling, in patients with a confirmed VVS diagnosis (157 subjects) and individuals without a history of syncope (161 subjects). We were able to identify the associations between VVS and alleles/genotypes *ADRA1A* rs1048101, *ADRB1* rs1801253, *ADORA2A* rs5751876, and *COMT* rs4680, as well as *NOS3* rs2070744 in biallelic combination with *COMT* rs4680. Thus, we are the first to observe, within a single study, the role of the genes that encode α- and β-adrenergic receptors, catechol-O-methyltransferase, adenosine receptors and nitric oxide synthase in VVS development. These findings demonstrate that the genes involved in neurohumoral signaling pathways contribute to the formation of a genetic susceptibility to VVS.

## 1. Introduction

Syncope occurs in more than 30% of the population [1]. The most common type of syncope is vasovagal syncope (VVS), which develops due to abnormal autonomic control of circulation, leading to a decrease in blood pressure (BP) and cerebral hypoperfusion. Despite a favorable life prognosis, VVS significantly reduces the quality of life and may lead to physical and mental injuries [2].

Although the mechanisms underlying VVS are not completely clear, there is no doubt that its development is influenced by many factors, among which hereditary predisposition holds an important position [3,4,5]. Identification of VVS-associated genetic factors can expand the understanding of the molecular basis of its pathogenesis, and thus contribute to the development of new prevention and treatment strategies.

In view of the available information, VVS development may be affected by genes, the products of which participate in the functioning of autonomic nervous system (ANS) and cardiovascular system (CVS). Consequently, genes that encode receptors, carrier proteins, and enzymes involved in the synthesis of neurotransmitters and vasoactive molecules were considered as possible VVS susceptibility loci. However, at the moment, data on the role of individual candidate genes in VVS are very inconsistent [3,4,5], which may be due to the small sample sizes, variations in criteria for VVS diagnosis, and insufficiently rigorous selection of control groups; these problems certainly need to be addressed.

ANS and CVS are partly controlled by neurohumoral regulation that triggers cascades of physiological reactions in the body, directly or indirectly affecting vascular tone, heart function, and, as a result, BP. The development of VVS is associated with a wide range of neurohumoral changes that influence the levels of epinephrine, norepinephrine, adenosine, nitric oxide, vasopressin, endothelin, and other vasoactive molecules [6].

In this study, we analyzed how VVS is associated with polymorphic variants in the *ADRA1A*, *ADRB1*, *HTR1A*, *ADORA2A*, *COMT*, and *NOS3* genes involved in neurohumoral signaling, using representative samples of patients with a confirmed VVS diagnosis and individuals without a history of syncope.

## 2. Materials and Methods

### 2.1. Subjects

One hundred and fifty-seven VVS patients from Chazov National Medical Research Center of Cardiology (mean age ± standard deviation (SD), 35.9 ± 15.7 years) were involved in the study. The sample included 64 males (mean age ± SD, 36.3 ± 16.0 years) and 93 females (mean age ± SD, 35.6 ± 15.6 years). The criteria for VVS diagnosis were based on a clinical survey, examination, and a tilt test (a long-term passive orthostatic test that provokes syncope) or a bicycle ergometer test performed according to a special protocol [7]. Individuals with loss of consciousness due to cardiac, neurological and/or metabolic reasons were excluded from the study.

The control group included 161 individuals (mean age ± SD, 38.6 ± 14.3 years) without a history of syncope, in particular 90 males (mean age ± SD, 37.5 ± 13.3 years) and 71 females (mean age ± SD, 40.1 ± 15.5 years). Based on the examination, 30 subjects were not diagnosed with any illnesses, and 131 patients were diagnosed with various cardiac arrhythmias (rare paroxysms of atrioventricular nodal reciprocating tachycardia, flutter, or atrial fibrillation) or transient hypertension. There were no signs of organic myocardial damage or other somatic or neurological diseases in individuals of the control group.

### 2.2. DNA Extraction and Genotyping

Genomic DNA was isolated from the peripheral blood using a commercial QIAamp DNA Blood Mini Kit (QIAGEN, Germany). Genotyping of polymorphic regions *ADRA1A* rs1048101, *ADRB1* rs1801253, *HTR1A* rs6295, *ADORA2A* rs5751876, *COMT* rs4680, and *NOS3* rs2070744 was performed using real-time PCR with commercial TaqMan^®^ SNP Genotyping assays (ThermoFisher Scientific, Waltham, MA, USA), according to the manufacturer’s instructions.

### 2.3. Statistical Analysis

Deviations in the observed genotype frequencies from the Hardy–Weinberg equilibrium were analyzed by χ^2^ test using Haploview 4.2 software (Broad Institute, Cambridge, MA, USA) [8]. Pairwise linkage disequilibrium of polymorphic regions on the same arm of a chromosome was also calculated using Haploview 4.2 software. The association of allele frequencies and allele/genotype carriage frequencies with VVS was analyzed by comparing allele frequencies and allele/genotype carriage frequencies in VVS patients and controls with the two-sided Fisher’s exact test using the GraphPad Instat software [9]. The multilocus analysis was applied to identify biallelic combinations, the carriages of which are associated with VVS, using the APSampler software [10,11]. The association of combinations with the disease was assessed using the criterion of a minimum set of alleles as a genetic risk factor, according to which any allele/genotype included in the identified combination is characterized by a lower significance of the association than the combination, and addition of any additional allele(s) to the combination does not increase the significance of its association. Differences in compared frequencies were considered significant if the *p*-value less than 0.05 and 95% confidence interval (CI) for odds ratio (OR) did not cross 1. The Bonferroni correction procedure was performed to correct for multiple hypothesis testing.

The logistic regression analysis and receiver operating characteristic (ROC) analysis were performed to assess prognostic significance of disease predictors using the GraphPad Prism v8.4.3 (GraphPad Software, San Diego, CA, USA).

## 3. Results

By applying the classical candidate gene approach for the association analysis, we selected polymorphic variants, located in genes that are involved in neurohumoral signal transduction pathways, which, in turn, are implicated in VVS development (Table 1).

There were no deviations from the Hardy–Weinberg equilibrium (*p* > 0.01) for any of the studied single nucleotide polymorphisms (SNPs), in either VVS patients or controls. No linkages between polymorphic regions of the *COMT* and *ADORA2A* genes located on the long arm of chromosome 22 were revealed (D′ < 1, LOD < 2); these results allow us to further consider SNPs rs4680 and rs5751876 as independent markers.

Table 2 shows the distribution of allele frequencies, as well as allele and genotype carriage frequencies, of the studied polymorphic variants in VVS patients and controls, for which a significant difference between groups was observed. Compared with the controls, VVS patients were characterized by higher frequencies of alleles *ADRA1A**G (*p* = 0.021, OR = 1.54, 95% CI: 1.07–2.22), *ADORA2A**C (*p* = 0.037, OR = 1.41, 95% CI: 1.03–1.94), and *COMT**G (*p* = 0.0043, OR = 1.6, 95% CI: 1.17–2.18). The comparison of allele and genotype carriage frequencies revealed that VVS patients more often carried *ADRA1A**G/G (*p* = 0.032, OR = 1.64, 95% CI: 1.05–2.56), *ADRB1**G (G/G + C/G) (*p* = 0.048, OR = 1.61, 95% CI: 1.02–2.55), *ADORA2A**C/C (*p* = 0.013, OR = 1.83, 95% CI: 1.14–2.93), and *COMT**G (G/G + A/G) (*p* = 0.0094, OR = 1.96, 95% CI: 1.2–3.22). However, it should be noted that the *p*_corr_ value after Bonferroni adjustment for multiple comparison remained significant only for the *COMT**G allele. There were no significant differences in allele frequencies and allele/genotype carriage frequencies between the VVS patients and healthy controls for polymorphic variants in *HTR1A* and *NOS3* genes.

To identify the contribution of the co-carriage of alleles and genotypes of the studied genes to VVS susceptibility, we performed multilocus analysis using APSampler software. We found combinations of alleles, the carriage of which was associated with the risk of VVS; these combinations were characterized by a higher level of significance and a larger effect size than their components (Figure 1). The predisposing biallelic combination of the *NOS3**T allele (the carriage of which, by itself, was not significantly associated with VVS) with the *COMT**G allele (associated with VVS by itself, *p* = 0.0094, OR = 1.96, 95% CI: 1.2–3.22) is characterized by *p* = 0.0013 and OR = 2.11, 95% CI: 1.31–3.37, i.e., it is almost by one order more significantly associated with the disease than the single *COMT**G allele (Figure 1A). Carriage of the protective biallelic combination *ADRA1A**A + *ADORA2A**T (*p* = 0.00083, OR = 0.45, 95% CI: 0.27–0.73) is associated with a lower risk of VVS and is characterized by an increase in the level of significance compared with carriage of *ADRA1A**A and *ADORA2A**T by themselves (*p* = 0.032 and 0.013, respectively) (Figure 1B).

We assessed the predictive efficacy of the identified genetic risk factors using the logistic regression method (Figure 2). The considered genetic risk factors for VVS were found to be poor classifiers; the area under the curve (AUC) did not reach 0.60 for either single alleles or the biallelic combination *COMT**G + *NOS3**T. In a composite model that includes allele/genotype carriage frequencies for genes *ADRA1A*, *ADRB1*, *ADORA2A*, and *COMT*, a satisfactory predictive efficacy (AUC = 0.64) is achieved. Replacing *COMT**G with the *COMT**G + *NOS3**T combination does not make the model more efficient.

## 4. Discussion

In order to study the genetic predisposition to VVS, we formed the following 2 groups that included more than 150 subjects each and met fairly strict criteria: “case”—a group of patients with a confirmed VVS diagnosis, and “control”—a group of individuals without syncope. The characteristics of these groups favorably distinguish our study from many others with similar objectives. Indeed, a systematic review [3] showed that for half of the polymorphic variants ever explored for VVS, the association analysis was performed in less than 100 patients. Many studies used comparison groups that did not allow the researchers to fully assess genetic susceptibility factors for VVS; about half of the studies compared the genotype frequencies in subgroups of VVS patients, formed on the basis of negative or positive tilt tests or the type of induced VVS [3].

Of the many genes known to be involved in the regulation of ANS and CVS functioning via neurohumoral signaling pathways, we selected *ADRA1A* and *ADRB1* genes that encode α1A- and β1-adrenergic receptors, respectively, as well as *HTR1A* and *ADORA2A* genes, encoding serotonin 1A and adenosine A2A receptors. These receptors belong to a large superfamily of G-protein-coupled receptors and, through coordinated action of various G-proteins, stimulate the intracellular signaling cascades to maintain body homeostasis. Activation of α1A-adrenergic receptors initiates contraction of smooth muscles that leads to the spasm of arterioles and an increase in BP; activation of β1-adrenergic receptors causes an increase in the strength and frequency of heart rate (HR) [21]. On the contrary, activation of serotonin 1A and adenosine A2A receptors leads to vasodilation, reduced BP, and decreased HR [22]. We also included in the panel the *COMT* gene encoding catechol-O-methyltransferase, which is involved in inactivation of ligands of α1A- and β1-adrenergic receptors, namely epinephrine and norepinephrine [21], and the *NOS3* gene encoding endothelial nitric oxide synthase, which is involved in the synthesis of one of the main vasodilators, nitric oxide NO [23]. Epinephrine and norepinephrine are involved in the activation of the latter enzyme in addition to other regulators [23,24].

We were able to identify associations between VVS and variants in *ADRA1A* rs1048101, *ADRB1* rs1801253, *ADORA2A* rs5751876, and *COMT* rs4680, as well as *NOS3* rs2070744 (as part of a biallelic combination with *COMT* rs4680). Thus, the analysis of allelic polymorphism revealed, for the first time within a single study, the importance for VVS development of the genes that encode α- and β-adrenoreceptors and catechol-O-methyltransferase, which mediate the effect of neurotransmitters on the ANS, and adenosine receptor and nitric oxide synthase genes, which mediate the effect of adenosine and nitric oxide on the CVS.

Our data on the association of the *ADRA1A* rs1048101*G allele and *G/G genotype with VVS are in agreement with the results reported in [25], while in other studies, no association between rs1048101 and VVS or the tilt test response was found [5,26,27].

The *ADRB1* rs1801253 has been mentioned in the largest number of VVS association studies; however, the results are contradictory [3]. In two studies, as was the case in ours, VVS patients were compared with individuals from the control group, but no association between SNP and VVS was found [5,28]. However, in other studies, this genetic variant was shown to be associated with tilt test response in VVS patients [29,30].

Again, in contradiction to our results, Saadjian et al. did not find any association of VVS with the alleles of *ADORA2A* when comparing patients with healthy individuals; however, the *ADORA2A**C/C genotype was found to be associated with a positive tilt test and a high frequency of syncope episodes [31]. In other studies, no association with either tilt test response or VVS itself were found [5,32].

The *COMT**G allele was found to be associated with VVS in our study. This allele was previously reported to be associated with VVS in males, whereas its protective effect was observed in females [5].

We did not find any association between VVS and variants of *HTR1A* rs6295 and *NOS3* rs4680. The influence of these polymorphisms in VVS had been considered only in [5], which reported an association between VVS and *HTR1A* rs6295 only in males and no association, as in our study, between VVS and *NOS3* rs4680. However, due to the use of multilocus analysis, which increases the statistical power of a study [33], we identified the biallelic combination of *NOS3* rs4680*T allele with the *COMT**G allele, carriage of which increases risk of VVS development.

Thus, for all the polymorphic regions for which we observed an association with VVS, there is some published data that are consistent with our findings. At the same time, there are even more studies with no association found for the same polymorphisms. Although we cannot exclude the possibility of false positive results, it seems more likely that due to the adequate formation of comparison groups and the use of multilocus analysis, we managed to obtain results that reflect the actual involvement of the studied genes in the development of VVS.

Taking into account the published data, we assessed a possible impact of the identified VVS-associated allelic variants on the disease development and found that their carriage may contribute in varying degrees to the main factors of VVS pathogenesis, including decreased vascular tone, reduced HR, and hypotension. Amino acid substitutions defined by variants of *ADRA1A* rs1048101 (Cys347Arg) and *ADRB1* rs1801253 (Gly389Arg) are located at the C-terminal regions of receptors and affect protein interactions with G-proteins and intracellular signal transduction [15,16]. The presence of Arg347 in the α-1 adrenoreceptor (*ADRA1A* rs1048101*G allele) may lead to faster internalization of the receptor and, as a result, to a decrease in the intracellular concentration of calcium ions, vasodilation, and a decrease in venous return, which increases the risk of lowering BP and VVS development [25]. Compared to the Arg389, the Gly389 variant in the β1-adrenergic receptor (*ADRB1* rs1801253*G allele) acts as a hypofunctional variant when overexpressed in vitro and in response to b-agonists in healthy volunteers [34,35,36], and may reduce the strength and rate of heart contractions, i.e., promote bradycardia. According to the published data, carriers of the *COMT* rs4680*G (Val158) allele are characterized by increased activity of catechol-O-methyltransferase [17]. This enzyme is the most important regulator of adrenoreceptor-mediated signal transduction, participating in the breakdown of catecholamines (in particular, epinephrine and norepinephrine), which largely controls the level of adrenoreceptor activation [37]. Increased activity of catechol-O-methyltransferase may decrease activation of adrenoreceptors and, as a result, reduce BP and HR. SNP rs2070744 is located in the promoter region of the *NOS3* gene; carriage of the rs2070744*T allele is associated with increased *NOS3* transcription [13] and a higher plasma NO level [14]. NO promotes vasodilation and reduces BP. SNP rs5751876 in the *ADORA2A* gene leads to a synonymous substitution. Although we did not find studies on the association between this polymorphic variant and the effect on the rate of heart contractions or vascular tone, carriage of the C allele has been shown to be associated with a reduced level of anxiety after caffeine consumption [18,19] and a lower incidence of panic attacks [20]. These data may be explained by the direct influence of this SNP or other SNPs within this gene that are in linkage disequilibrium with it [20].

In general, our data suggest that the development of VVS undoubtedly has a genetic component and, in most cases, is affected by small impacts of polymorphic variants of many genes. We have shown the association of the genes involved in neurohumoral signaling with VVS, although the predictive power of their joint contribution is quite low (the AUC value for a composite model is 0.64). The limitation of this study is the absence of a validation group, which would certainly strengthen the study, but nonetheless, in this field, the sample size is quite good. The search for other genetic risk factors for VVS should continue at pace.

## Figures and Tables

**Figure 1 genes-13-01653-f001:**
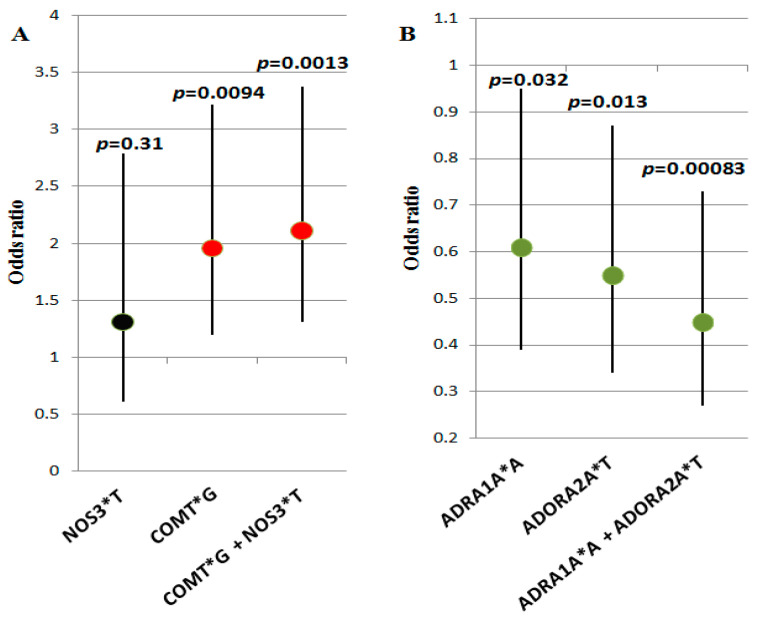
Combinations of alleles, the carriage of which is associated with the risk of VVS: (**A**) predisposing combination *COMT**G + *NOS3**T and its alleles; (**B**) protective combination *ADRA1A**A + *ADORA2A**T and its alleles. Odds ratio values (dots) and confidence intervals of genetic variants (vertical segments) are presented graphically; *p*-values are denoted on the top. Black dots denote insignificant variants, red dots denote predisposing variants, and green dots denote protective variants.

**Figure 2 genes-13-01653-f002:**
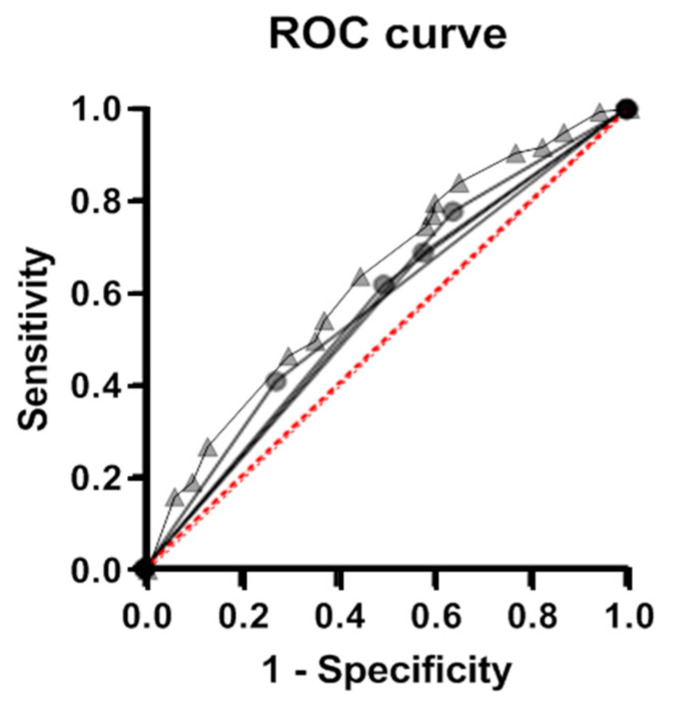
ROC analysis of the effectiveness of the models performed for the identified genetic variants of individual risk of VVS and for a composite model. Efficacy of the classification of individuals using models based on the carriage of individual genetic markers (*ADRA1A*, *ADRB1*, *ADORA2A*, and *COMT* gene variants, circles) and a model allowing for carriage of variants in all four genes (composite genetic marker, triangles).

**Table 1 genes-13-01653-t001:** Genes and single nucleotide polymorphisms (SNPs) selected for the association analysis.

Gene and Its Chromosomal Localization	rs ID and Polymorphism (Amino Acid Substitution)	Encoded Protein	Known (Putative) Effects on Level/Activity of the Product	Bioactive Molecules Mediating Neurohumoral Signal Transduction Pathways
*HTR1A* 5q12.3	rs6295 −1019 G>C	Serotonin 1A receptor	SNP blocks the function of specific repressors Hes1, Hes5 and Deaf1, resulting in increased 5-HT1A autoreceptor expression in animal models and humans [12].	Serotonin
*NOS3* (*eNOS*) 7q36.1	rs2070744 −786 T>C	Endothelial nitric oxide synthase	SNP affects the *NOS3* transcription [13] and the plasma NO level [14].	Nitric oxide
*ADRA1A* 8p21.2	rs1048101 1039 A>G (Cys347Arg)	α 1A-adrenergic receptor	Amino acid substitution Cys347Arg affects receptor interactions with G-proteins and intracellular signal transduction [15].	Epinephrine, norepinephrine
*ADRB1* 10q25.3	rs1801253 1165 G>C (Gly389Arg)	β 1-adrenergic receptor	Amino acid substitution Gly389Arg affects receptor interactions with G-proteins and intracellular signal transduction [16].	Epinephrine, norepinephrine
*COMT* 22q11.21	rs4680 472 G>A (Val158Met)	Catechol-O- methyltransferase	Amino acid substitution Val158Met affects the activity of catechol-O-methyltransferase [17].	Epinephrine, norepinephrine, dopamine
*ADORA2A* 22q11.23	rs5751876 1083 C>T (Tyr361Tyr)	Adenosine A2A receptor	SNP is associated with a level of anxiety after caffeine consumption [18,19] and incidence of panic attacks [20].	Adenosine

**Table 2 genes-13-01653-t002:** Allele frequencies and allele/genotype carriage frequencies for polymorphic variants in genes associated with VVS.

Alleles and Genotypes	Patients, *n* (%) N = 157	Controls, *n* (%) N = 161	*p* Value (*p_corr_*)	OR (95% CI) for Significant Differences
***ADRA1A* rs1048101**				
Allele frequency				
**A**	**64 (20)**	**91 (28)**	**0.021**	**0.65 (0.45–0.94)**
**G**	**250 (80)**	**231 (72)**	**0.021**	**1.54 (1.07–2.22)**
Allele and genotype carriage frequency				
**A (A/A + A/G)**	**60 (38)**	**81(50)**	**0.032**	**0.61 (0.39–0.95)**
G (G/G + A/G)	153 (98)	151 (94)	NS	–
A/A	4 (2)	10 (6)	NS	–
A/G	56 (36)	71 (44)	NS	–
**G/G**	**97 (62)**	**80 (50)**	**0.032**	**1.64 (1.05–2.56)**
***ADRB1* rs1801253**				
Allele frequency				
C	176 (56)	204 (63)	NS	–
G	138 (44)	118 (37)	NS	–
Allele and genotype carriage frequency				
C (C/C + C/G)	127 (81)	136 (84)	NS	–
**G (G/G + C/G)**	**108 (68)**	**93 (58)**	**0.048**	**1.61 (1.02–2.55)**
**C/C**	**49 (32)**	**68 (42)**	**0.048**	**0.62 (0.4–0.98)**
C/G	78 (49)	68 (42)	NS	–
G/G	30 (19)	25 (16)	NS	–
***ADORA2A* rs5751876**				
Allele frequency				
**C**	**195 (62)**	**173 (54)**	**0.037**	**1.41 (1.03–1.94)**
**T**	**119 (38)**	**149 (46)**	**0.037**	**0.7 (0.51–0.97)**
Allele and genotype carriage frequency				
C (C/C + C/T)	131 (84)	129 (81)	NS	
**T (T/T + C/T)**	**93 (59)**	**117 (70)**	**0.013**	**0.55 (0.34–0.87)**
**C/C**	**64 (41)**	**44 (30)**	**0.013**	**1.83 (1.14–2.93)**
C/T	67 (43)	85 (51)	NS	
T/T	26 (16)	32 (19)	NS	
***COMT* rs4680**				
Allele frequency				
**A**	**140 (45)**	**181 (56)**	**0.0043 (0.026)**	**0.63 (0.46–0.86)**
**G**	**174 (55)**	**141 (44)**	**0.0043 (0.026)**	**1.6 (1.17–2.18)**
Allele and genotype carriage frequency				
A (A/A + A/G)	105 (68)	123 (76)	NS	–
**G (G/G + A/G)**	**122 (78)**	**103 (64)**	**0.0094**	**1.96 (1.2–3.22)**
**A/A**	**35 (22)**	**58 (36)**	**0.0094**	**0.51 (0.31–0.84)**
A/G	70 (46)	65 (40)	NS	–
G/G	52 (32)	38 (24)	NS	–

Significant differences are shown in bold; NS—not significant (*p* > 0.05); OR (95% CI) is given only for *p* < 0.05.

## Data Availability

Not applicable.

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
