# Peer review of "Vasovagal Syncope Is Associated with Variants in Genes Involved in Neurohumoral Signaling Pathways"

_genes, 2022, doi:10.3390/genes13091653_

Round 1
Reviewer 1 Report
The manuscript titled "Vasovagal syncope is associated with variants in genes involved in neurohumoral signaling pathways." by Boris et al. is a well-organized study and a relevant observation in the cardiovascular biology. I would recommend the abstract be made better by language correction. And minor edits in language throughout.
Best of luck
Author Response
Dear Reviewer,
Thank you for the appreciation of our manuscript. Following your recommendations, we asked for assistance from an experienced translator and improved English in Abstract and throughout the text (all changes are shown).
Reviewer 2 Report
In this study 157 subjects with vasovagal syncope were compared with hundred 61 control subjects who did not have vasovagal syncope, comparing six genetic polymorphisms seen in genes linked to vasovagal syncope previously. The study has some positive results suggesting that at least some of these polymorphisms are significant. The paper is set out clearly and while I am not a statistician the statistics is also explained to my satisfaction and the English is clear and understandable.
A difficulty in the area is that the results disagree with some previous work while agreeing with others making one somewhat suspicious that some of this may be background noise. The paper would be strengthened by having a second validation cohort, but nonetheless, in this field the sample size is quite good.
It would be helpful to be a bit more clear about why each of the polymorphisms were selected. This could be done for example by enlarging table 1 a little to include the references supporting the inclusion of each polymorphism.
While I would certainly not make this an essential addition, the authors may consider adding a summary diagram to include the biochemical or physiological mechanisms by which the polymorphisms putatively makes vasovagal syncope more or less likely.
Author Response
Dear Reviewer,
Thank you for the appreciation of our manuscript and for the helpful recommendations.
A difficulty in the area is that the results disagree with some previous work while agreeing with others making one somewhat suspicious that some of this may be background noise. The paper would be strengthened by having a second validation cohort, but nonetheless, in this field the sample size is quite good.
We do agree with your comment and realize that a second validation cohort will strengthen the study. We addressed this limitation in the last paragraph of Discussion:
“The limitation of this study is the absence of a validation group, which would definitely strengthen the study, but nonetheless, in this field the sample size is quite good.”
It would be helpful to be a bit more clear about why each of the polymorphisms were selected. This could be done for example by enlarging table 1 a little to include the references supporting the inclusion of each polymorphism. While I would certainly not make this an essential addition, the authors may consider adding a summary diagram to include the biochemical or physiological mechanisms by which the polymorphisms putatively makes vasovagal syncope more or less likely.
Thank you for these recommendations. We now modified Table 1 by adding new column “Known (putative) effects on level/activity of the product”, which includes the references supporting the selection of each polymorphism for the study; we also merged two columns “Gene” and “Chromosomal localization” as well as two columns “rs ID” and “Polymorphism (amino acid substitution)” in order to present the table in a more structured fashion. We also took the liberty not to include a summary diagram in the manuscript since we now added the information on biochemical or physiological mechanisms by which selected polymorphisms putatively involved in VVS in the new column of Table 1, in addition to previous text in the Discussion (paragraph 2).